# Dual-Signal Feature Spaces Map Protein Subcellular Locations Based on Immunohistochemistry Image and Protein Sequence

**DOI:** 10.3390/s23229014

**Published:** 2023-11-07

**Authors:** Kai Zou, Simeng Wang, Ziqian Wang, Hongliang Zou, Fan Yang

**Affiliations:** 1School of Communications and Electronics, Jiangxi Science and Technology Normal University, Nanchang 330038, China; 2School of Computer Science and Technology, Huazhong University of Science and Technology, Wuhan 430074, China; 3Artificial Intelligence and Bioinformation Cognition Laboratory, Jiangxi Science and Technology Normal University, Nanchang 330038, China

**Keywords:** dual signal, benchmark database, discriminative feature operators, protein subcellular location prediction

## Abstract

Protein is one of the primary biochemical macromolecular regulators in the compartmental cellular structure, and the subcellular locations of proteins can therefore provide information on the function of subcellular structures and physiological environments. Recently, data-driven systems have been developed to predict the subcellular location of proteins based on protein sequence, immunohistochemistry (IHC) images, or immunofluorescence (IF) images. However, the research on the fusion of multiple protein signals has received little attention. In this study, we developed a dual-signal computational protocol by incorporating IHC images into protein sequences to learn protein subcellular localization. Three major steps can be summarized as follows in this protocol: first, a benchmark database that includes 281 proteins sorted out from 4722 proteins of the Human Protein Atlas (HPA) and Swiss-Prot database, which is involved in the endoplasmic reticulum (ER), Golgi apparatus, cytosol, and nucleoplasm; second, discriminative feature operators were first employed to quantitate protein image-sequence samples that include IHC images and protein sequence; finally, the feature subspace of different protein signals is absorbed to construct multiple sub-classifiers via dimensionality reduction and binary relevance (BR), and multiple confidence derived from multiple sub-classifiers is adopted to decide subcellular location by the centralized voting mechanism at the decision layer. The experimental results indicated that the dual-signal model embedded IHC images and protein sequences outperformed the single-signal models with accuracy, precision, and recall of 75.41%, 80.38%, and 74.38%, respectively. It is enlightening for further research on protein subcellular location prediction under multi-signal fusion of protein.

## 1. Introduction

Subcellular proteomics emerged as a unique area of proteomics in eukaryotic cells with the advent of organelle separation technology and proteomics technology, which are conducive to analyzing protein composition, determining the role of subcellular structures, and researching physiological or pathological environments [1,2]. In compartmentalized cellular structures, proteins appear in the right subcellular neighborhoods and execute biochemical processes or biological functions [2]. Moreover, aberrant locations of proteins have been linked to many pathological conditions, such as breast cancer, and it provides clues for comprehending protein function and designing cancer drug-targets [3,4]. Therefore, predicting protein subcellular location has attracted wide attention, and the automated systems would address the shortcomings of being time- and labor-consuming compared with various traditional wet-experiment approaches [5].

Automated classification models for predicting protein subcellular locations have been around for a while. They were divided into two groups based on various protein patterns: one is based on protein sequence assembled by 1D amino acid; the other is directed to 2D microscopic images. Both categories work on the workflow of three-step pipelines: data preprocessing, extracting discriminative features that distinguish protein subcellular location, and applying an effective classifier to fit data distribution [6].

One major pipeline in the sequence-based method is that discriminative features were extracted to explore the correlation of 20 natural amino acids between similar protein sequences, including hand-crafted features based on traditional statistical algorithms and abstract features from representation learning. Some feature extraction methods based on the composition of amino acids were defined to express correlation between amino acids: such as k-mer frequencies, Position-Specific Scoring Matrix (PSSM), and Pseudo-Amino Acid Composition (PseAAC) [7,8,9]. On the other hand, the peptide-based and annotation-based features were also introduced to search for homologous proteins or fitting protein subcellular location distributions, such as sorting signals, functional domains, sequence motifs, and the gene ontology (GO) terms [10,11,12,13,14]. Benefitting from previous works, subsequent scientists proposed more effective feature extraction algorithms. Specifically, the functional domain information and the sequential evolution information were fused to construct ensemble predictor by Shen et al. [15]. In Guo et al., a fusion feature space consisting of PseAAC and PSSM was fed into a feature–label matrix to recognize multi-label protein samples [16]. A novel representation protocol named Hidden Correlation Modeling (HCM) was utilized to create more compact features by exploring hidden correlations between GO annotation terms [17]. The dipeptide information with the blank space, coupled with the gapped k-mer information, was calculated to describe protein properties by quad-tree [18]. In addition, as deep learning advanced, a sequence-based deep-learning approach was proposed, which combines text information from amino acid components and deep-learning models. For instance, attention mechanism and a Convolution Neural Network (CNN) were employed to identify motif information of protein sequence in Armenteros et al.; then, the Recurrent Neural Networks (RNNs) can be viewed as a trainable encoder to map a convincing category space [19]. In Liao et al., bidirectional Long–Short-Term Memory (LSTM) and CNN were used to refine amino acid composition sequences and evolution matrices of proteins; next, the outputs from two deep-learning models mentioned previously were concatenated and flattened to one-dimensional classification array [20]. Following that, systems built using representation learning and post-process hybrid classifier algorithms have been created to enhance the bias of positive samples [21,22]. The above sequence-based models were successful in acting on various scenarios and valuable in laying the foundation for subsequent research, while 2D microscope images steadily gain attention due to their objectivity and impressive interpretation.

Two-dimensional microscopic images have grown in popularity with the advent of microscopic technology since they can show the spatial distribution of proteins in various tissues [23]. Similar to the research protocol of sequence-based models, image-based models based on digital image-processing technologies and machine-learning algorithms have been developed to analyze subcellular patterns of protein. At present, immunohistochemistry (IHC) and immunofluorescence (IF) images have been used as mainstream research objects. The Subcellular Location Features (SLFs) included morphological features, Zernike moment features, Haralick texture features, and wavelet features, which were employed to quantitatively express statistical characteristics of protein subcellular location distribution [24,25]. It represents digital information of a global view on microscope images, such as Zernike moment features showing translation rotation invariance, Haralick features following Gray-Level Co-occurrence Matrix (GLCM) in various orientations, and DNA spatial distribution conveys overlapping and distance in protein and nuclear [26,27]. In addition to this, Local Binary Pattern (LBP), Local Ternary Pattern (LTP), Local Quinary Pattern (LQP), and other local operators portray local texture, edge, and other structure information [28,29,30]. Additionally, a monogenic signal based on the Log-Gabor filter and intensity coding strategy in frequency space was adopted to avoid the sparse problem of protein subcellular image [31]. Some deep features from deep-learning models were extracted to depict protein images as deep-learning flourishes in natural images, such as abstract features, abstract morphology, and region contour. To improve supervisory efficacy of protein microscopic images, feature fusion that links shallow and abstract features was implemented [32,33,34]. And further down the line, plentiful deep-learning algorithms, such as image attention mechanism, transfer learning, and other classical algorithms, were introduced to analyze subcellular location patterns of proteins [35,36,37,38,39]. Previous studies have made great contributions based on various protein patterns, such as protein sequence, IHC images, and IF images. However, few multi-signal fusion methods received attention due to label inconsistencies and differences in quantitative representations of different protein patterns [40,41,42].

In this paper, we reported a dual-signal fusion system for predicting protein subcellular location by incorporating IHC images into protein sequences, and some key points can be summarized as follows. An important point of all is that a benchmark dataset with a common subcellular location of protein was collected and generalized as the experimental objects from the HPA and Swiss-Prot database. Then, several classical feature operators were employed to quantify IHC images and protein sequences, and abstract features derived from deep-learning models were extracted to depict IHC images. Finally, feature spaces from the mentioned dual signal were applied to build multiple sub-classifiers, and the decision label was generated using a centralized voting mechanism. From the experimental results, the dual-signal fusion method based on protein sequence and IHC images can reach 75.41%, 80.38%, and 74.38% in accuracy, precision, and recall, respectively. It outperforms other single-signal models. Furthermore, it also serves as a reasonable reference for subsequent multi-signal studies.

## 2. Materials and Methods

### 2.1. The Benchmark Dataset

Appropriate datasets in the machine-learning models would yield significant influence. In this work, the Human Protein Atlas (https://www.proteinatlas.org/, accessed on 4 May 2022) database was adopted to collect IHC images under tissue cell types according to data criteria. Specifically, there are two criteria for filtering IHC images, i.e., the protein expression level presents the staining level of the protein channel, and the reliability score judged the reliability level for the subcellular localization annotation of protein. Beyond that, the Swiss-Prot database in Universal Protein Resource (https://www.uniprot.org/, UniProt, accessed on 11 May 2022) was used to collect the protein sequence dataset. The 4722 proteins involved in nine major subcellular locations contained centrosome, endoplasmic reticulum (ER), Golgi apparatus (Golgi), mitochondria, nucleoli, vesicles, cytosol, nucleoplasm, and plasma membrane, initially selected in HPA. Then, a benchmark dataset described as high quality and with the same label was collected from the two databases according to the following steps.

First, a collection with 287 paired samples was captured in line with three rules. The first is that the protein sequence location in Swiss-Prot should keep the same labels as the microscopic images’ labels in HPA; the second is to capture IHC images that describe protein status with high staining levels at the tissue level; the third is that the subcellular location of all proteins should be verified via biochemistry wet-experiment. According to the above conditions, 287 protein sequences from Swiss-Prot and HPA were sorted out, which involved four protein subcellular locations, including ER, Golgi, cytosol, and nucleoplasm.

Second, homologous protein sequences were removed, and an appropriate number of IHC image datasets were matched. In the 287 protein sequences involved, some homologous protein sequences were deleted to reduce sequence redundancy and enhance the independence of non-homologous protein sequences via CD-HIT (V4.6) software [43], in which the sequence identity cut-off was set to 0.9. To facilitate feature extraction of sequences in subsequent stages, the sequences with a length between 50 and 5000 are reserved. After processing, the remaining 281 protein sequences became the final workable dataset. At the same time, one image was randomly selected from IHC images of protein sequence expression at the tissue level to match the protein sequence. Therefore, the specific protein sequence dataset and IHC images dataset are shown in Table 1.

### 2.2. Analyzing Subcellular Patterns of Protein from Protein Sequence

The 281 protein sequences underwent redundancy elimination and length constraint by CD-HIT software, which is conducive to enhancing independence between non-homologous protein sequences and improving attribute-class differences for describing multiple feature information. The CD-HIT is a greedy algorithm. First, the longest of all protein sequences was considered the clustered representation. Then, common word counting is used to describe the similarity and redundancy of the remaining protein sequences.

In the protein sequences, some discriminative statistical operators were employed to quantify statistical properties of residues, as shown in Figure 1. The first type, the Position-Specific Scoring Matrix-based feature generator for machine learning (POSSUM), a versatile universal toolkit with an online web server for generating PSSM-based feature descriptors, was employed to obtain evolutionary information on protein sequences [44]. Herein, 18 types of PSSM-based feature descriptors derived from POSSUM were collected to express numerical representation of protein sequences. The iteration and E-value are 3 and 0.001, respectively, in parametric setting. Most of them are gained via matrix transformations, and the $E_{i,j}$ is the probability that an amino acid at the ith position of the sequence has mutated to the jth amino acid during evolution. The second type, a web server named PseAAC, was obtained to present sequence in a discrete mode, and not lose amino acid sequence-order information [9,45,46,47,48,49]. In this toolkit, the PseAA mode was set as Dipeptide-composition, so 420-dimension features were collected in this part, in which the first 20 components are conventional amino acid composition, and the last 400 components are the fractions of 400 dipeptides, such as AA, AC, and AD. The dep(q) is the total number of qth dipeptides in Figure 1. Moreover, protein sequences are synthesized through biochemical reactions between amino acids. Amino acid residues with physicochemical (PC) properties have an important impact on the function of forming proteins, so the PC properties of amino acids are introduced to describe protein sequences. The Pearson Correlation Coefficient (PCC) was used to calculate a 3675-dimensional vector for a protein sequence that was described by 50 PC properties [50]. It also expresses a correlation between different PC properties, and $pc_{l}^{p}$ is the pth PC property of the lth amino acid residue in the sequence.

After the above feature extraction, the feature space of the protein sequence was constructed to fit the sub-classifier. To avoid feature redundancy and model overfitting caused by too high dimension, the Least Absolute Shrinkage and Selection Operator (LASSO) was adopted in protein sequence features space [51]. The selected feature subspaces were fed into binary relevance (BR) for identifying the subcellular location of protein sequence [52].

### 2.3. Classifying Protein Subcellular Location Based on Shallow and Abstract Features of IHC Images

Some image-based models have been demonstrated to be advanced in predicting protein subcellular location by extracting shallow and abstract features [32,33,34,35,36]. In this work, IHC images from HPA were collected to predict subcellular locations, and the protocols can be summarized in Figure 1. First, getting protein-target regions from IHC images; second, extracting shallow and abstract features to qualify IHC images; finally, the fusion features were put into Stepwise Discriminant Analysis (SDA) to reduce the dimensions, and the derived subset features were fed into BR to fit category space of samples. The details are covered in the following sections.

#### 2.3.1. Focusing on Protein-Target Regions by Preprocessing

There are 281 IHC images that meet the staining level and match 281 protein sequences. In order to improve information richness and focus on the protein-target regions of IHC images, the 15 protein channel patch images were split from one RGB IHC image following the steps presented in Figure 1. Step 1, an empirical threshold value of 13 was adopted to filter out images with poor quality due to artificial staining; Step 2, the Linear Spectral Separation (LIN) based on color transforming theory was employed to separate the protein and DNA channel from IHC images [24]; Step 3, the protein channel with 3000 × 3000 resolution was split by a sliding window with 512 × 512, and the first 15 effective protein channel regions were inhaled as experimental objects to represent RGB IHC images. After the mentioned process, 15 protein channel patch images were adopted to extract shallow and abstract features.

#### 2.3.2. Quantifying IHC Images with Shallow and Abstract Features

In the protein channel patch images, the shallow and deep features were extracted from the statistical methods and deep-learning models to express the attributes of protein regions, as shown in Figure 1.

First, the shallow feature composed of Haralick and LBP features was extracted from protein channel patch images. The Haralick feature is a statistical feature based on discrete wavelet transform under GLCM. Protein channel patch images were scaled to 32 gray levels and decomposed down to 8 levels by discrete wavelet transform using the Daubechies filter [24]. Then, three-set 27-dimensional Haralick features were acquired from the horizontal, vertical, and diagonal reconstructed image, and 26-dimensional Haralick features were obtained from original protein channel patch image. Thus, one patch image was described by 674-dimension (3 × 27 × 8 + 26 = 674) Haralick features. The LBP feature depicted local texture, edges, and flat micropatterns by histogram statistics [27,53]. In the nine-grid mask of unit radius, the central pixel performs a Boolean operation with the adjacent pixels, and the 8-bit binary value is converted to a decimal value. After that, the frequency of each pixel level is counted as LBP features, so 256-dimensional statistical features are obtained [28].

Second, the abstract features were gained by feeding 15 protein channel patch images into different layers of the deep-learning model, and the average value of 15 vectors was calculated to represent the original image. A CNN composed of residual units and interactive pairwise attention modules was employed to extract abstract features [54,55]. Different from shallow features, abstract features can present the abstract information of images in a high-dimension space, in which feature maps undergo high-dimensional space transformation and numerous nonlinear functions in the deep-learning model. The feature maps from different layers of deep-learning models were extracted to present local morphological features and nondescript digital image information, which shows rich representation information and robust discriminative power [33,34,35]. Therefore, the Concatenat_3 (C3) with 1024-dimension and Global Average Pooling (GAP) layer with 1024-dimension in the CNN were acquired to depict protein channel patch images.

#### 2.3.3. Improving Performance of Multiple Classifiers by Centralized Voting Mechanism

In this paper, a sample containing a protein sequence and one IHC image is described via different discriminative quantization methods. Therefore, multiple sub-classifiers were produced after the above process. To further obtain the final output results, the centralized voting mechanism was employed to determine the subcellular location of protein. Specifically, the confidence that each protein belongs to four subcellular locations was evaluated by multiple sub-classifiers; then, the most reliable location derived from selected sub-classifiers would participate in the vote; finally, the subcellular location with the most votes was viewed as the decision result.

## 3. Results

According to the mentioned methods, two protein signals are represented via multiple discriminative feature operators. The entire experimental process can therefore be divided into three parts: the experimental results of protein sequences in statistical features, the experimental results of IHC images described by shallow and abstract features, and the decision performance for multiple classifier integration. Beyond that, the five-fold cross-validation strategy is utilized to produce more general experimental results.

### 3.1. The Shallow Features of Amino Acid Effectively Act on Protein Subcellular Location

The primary protein structure, which resembles the natural language, shows the chain structure of the protein sequence through the condensation and dehydration of various amino acids. Therefore, in describing the properties of protein sequences, 18 PSSM-based features are employed to characterize amino acid evolution, PseAAC features with 420-dimension are adopted to describe the properties of pseudo amino acids, and PC features with 3675-dimension are used to describe the physicochemical properties and similarity of amino acids. Moreover, to weaken the redundancy of features and avoid overfitting problems caused by excessively high feature dimensions, subset features were selected from original features by LASSO. Next, the subset features were fitted to a high-dimensional category space by the BR classifier with the Radial Basis Function (RBF) kernel function. In this section, 281 protein sequences were subjected to five-fold cross-validation, and the average value of five experimental results is taken as the output result. After the mentioned process, two graphs show the performance of individual features and concatenated features in Figure 2 and Figure 3.

It can be seen from Figure 2 that DP-PSSM (Directional Property PSSM) achieves the best performance among mentioned 20 individual features. Here, the 18 different PSSM-based features express protein specificity according to a variety of calculation methods, and the DP-PSSM is described to calculate the distance similarity between amino acids [56]. It can be seen from the results that DP-PSSM can achieve 55.51%, 55.83%, and 53.30% in accuracy, precision, and recall, respectively, which is 24.6%, 30.9%, and 24.62% higher than the worst result in the above three evaluation indicators. Therefore, DP-PSSM was selected as a partial of fusion feature space, and PseAAC features and PC features are also involved in pairwise concatenate. Four combinations of feature fusion schemes are shown in Figure 3. The feature space connected DP-PSSM with PC features, outperforms other cases, and can achieve 55.72%, 54.72%, and 53.89% in accuracy, precision, and recall, respectively. The experimental performance improved significantly in the concatenated feature space containing PseAAC features by comparing the results of only the PseAAC feature. Compared to experimental results before concatenating, the best results are only slightly floating on the original features space. Therefore, the representatives of the mentioned experimental results in the decision layer are the sample probabilities derived from connecting DP-PSSM features with PC features.

### 3.2. IHC Images Perform Better in Identifying Protein Subcellular Location Than Protein Sequence

The image-based system on protein subcellular location and protein-target regions of IHC patch images are conducive to weakening blank areas in the images and increasing the receptive field of feature operators [32,36]. Therefore, 15 protein channel patch images with 512 × 512 derived from original RGB IHC images with 3000 × 3000 are absorbed to extract shallow and abstract features to describe the original image. Among shallow features, the experimental results are presented in Figure 4. About abstract features, feature maps at different depths of CNN were extracted to present multiple digital image characteristics, the result can be seen in Figure 5. The C3 represents the output of the Concatenat_3 layer, and GAP represents the output of the global average pooling layer. The SDA was employed to decrease feature dimensions to avoid dimension curse and overfitting problems. The individual features of 15 protein channel patch images were averaged according to feature dimensions to represent an original sample.

Haralick features were calculated by discrete wavelet transform using db1-db10 Daubechies filters. It can be seen from Figure 4a that the Haralick feature involving 10 Daubechies filters is relatively stable in the overall change trend, and the overall performance falls within a limited range. After concatenating LBP features, a similar trend is shown in Figure 4b. The discrepancies in the sample distribution of IHC images and the properties of shallow features can account for these experimental results. First, proteins can be expressed in different tissues and presented as IHC images at various tissue levels. Therefore, the sample distribution of IHC images is affected via protein sample diversity and differences in protein regions at the tissue level. Second, Haralick features were used to describe statistical information from the global view of the IHC image, such as inertia, entropy, and energy. The LBP features were utilized to express local texture statistical information by using the histogram. Thus, basic image characteristics were described at the shallow features; limited specificity and weak discrimination were prone to unimpressive performance. The experimental results show that the best performance evaluation indicators appear in db8 no matter whether before or after concatenating.

Next, the fusion features that include feature maps at different depths of CNN and shallow features were applied; the experimental results are shown in Figure 5. Compared with shallow features, the performance of abstract features has been greatly improved. Moreover, the best performance can reach 72.39%, 75.48%, and 71.21% in accuracy, precision, and recall, respectively, after connecting GAP features, Haralick features with db8 Daubechies filters, and LBP features. The protein channel patch images, coupled with multi-view abstract features from CNN, lead to good performance. First, protein channel patch images with larger protein-target regions were transmitted in CNN, so that the characteristic properties of protein-target regions can be effectively captured. Second, protein channel patch images pass through some nonlinear mapping functions in CNN, and abstract representation is gradually revealed from shallow to deep layers, so the discrimination and robustness of feature maps are significantly improved. Among the above experimental results, some sub-classifiers were adopted in the decision layer of the model according to the above experimental results.

### 3.3. Superior Performance Obtained with the Centralized Voting Mechanism

So far, according to dual signals composed of protein sequence and IHC images, each protein sample has been digitally characterized from various features, so multiple sub-classifiers have been established. Therefore, to produce a final result, the centralized voting mechanism was employed to assemble multiple confidence of some predictors with better performance to obtain predicted labels. Here, five classifiers were adopted, i.e., the first is based on DP-PSSM and PC features of protein sequence; the second is based on C3 features of IHC images; the third is based on GAP features of IHC images; the fourth is based on C3, Haralick, and LBP features of IHC images; and the fifth is based on GAP, Haralick, and LBP features of IHC images; the results are shown in Figure 6. It can be seen that the performance has been effectively improved by integrating experimental results of the five-fold cross-validation of multiple sub-classifiers. As shown in Figure 6a, the best result can reach 85.96%, 85.23%, and 85.42% in accuracy, precision, and recall, respectively, and the overall average result can reach 75.41%, 80.38%, and 74.38% in the three evaluation indicators mentioned. To intuitively understand the prediction performance between different classes, confusion matrix visualization is employed to show how each class is predicted as other classes, as shown in Figure 6c–g. From the specific situation of confusion matrix, ER achieved the best 85.19% accuracy among four categories, followed by cytosol, nucleoplasm, and Golgi apparatus. In addition, a large number of protein samples were more biased to be predicted as ER, especially Golgi apparatus. Furthermore, a box plot is drawn to express the prediction performance fluctuation of different categories in five-fold cross-validation, as shown in Figure 6b. The results showed that experimental results of cytosol had the biggest fluctuant range, and nucleoplasm was the smallest.

## 4. Discussion

The experimental results from multiple sub-classifiers obtained can effectively improve the assessment indicators of the model by the centralized voting mechanism. It is undeniable that the results of five-fold cross-validation have fluctuated violently in the existing sample category distribution, according to the box plot in Figure 6b. The primary cause is the small data volume and imbalanced category distribution. Therefore, the Leave-One-Out (LOO) cross-validation is utilized to further illustrate modal performance to release the issue of unstable performance produced by the imbalanced dataset. Specifically, 280 out of 281 samples are selected as the training dataset in each fold experiment, and the remaining one sample is viewed as the test dataset. Here, five feature spaces were adopted, i.e., the first is based on C3 features of IHC images; the second is based on GAP features of IHC images; the third is based on C3, Haralick, and LBP features of IHC images; the fourth is based on GAP, Haralick, and LBP features of IHC images; and the fifth is based on DP-PSSM and PC features of protein sequence; the results are shown in Figure 7. It clearly expresses that the experimental results have significantly improved compared with five-fold cross-validation, and the best performance reached 84.70%, 83.31%, and 86.63% in accuracy, precision, and recall, respectively. Some information can be summarized as follows: First, experimental results of five predictors based on LOO were all significantly improved compared with the five-fold cross-validation; the main reason is that the category fitting ability of these classifiers was enhanced due to the increase in the number of training samples. Second, 280 out of 281 samples are utilized as training datasets, and only one sample in a category was viewed as the test dataset. It facilitates the representational ability of the model to fit the data distribution in maximizing the extent, which is conducive to expressing confidence in the test sample in the distribution space. On the other hand, although the experimental performance of the five classifiers has been significantly improved, the cost of time consumption is obvious. The main reason is that the single completed round required the completion of 281 model fitting procedures.

## 5. Conclusions

In this study, we developed a dual-signal model composed of protein sequences and IHC images to analyze protein subcellular location patterns. One contribution of this work is that a benchmark dataset with the same protein subcellular location label was collected from the HPA and Swiss-Prot databases and provides a dataset basis for following multi-signal studies of protein subcellular location. Additionally, according to different protein presentation patterns, corresponding feature extraction operators were employed to quantify and characterize protein samples. Specifically, 18 PSSM-based features, PseAAC features, and PC features were adopted to represent protein sequences. Moreover, Haralick features, LBP features based on statistical methods, and abstract features derived from the different depths of CNN were adopted to quantify IHC images. Specially, due to the quantitative differences of different protein signals, LASSO and SDA were applied to protein sequences and IHC images to reduce the feature dimensions to avoid dimension curse and overfitting. Finally, the output confidence of multiple BR sub-classifiers is absorbed to output confidence of protein at the subcellular location, and the centralized voting mechanism was employed to obtain the decision result. The experimental results show that the prediction performance of the dual-signal model based on protein sequence and IHC images can reach 75.41%, 80.38%, and 74.38% in accuracy, precision, and recall, respectively. It proved that the dual-signal model is advanced by incorporating IHC images into protein sequences. However, there is a clear data imbalance in the benchmark dataset from the Swiss-Prot and HPA databases, thus data equalization processing and discriminative representation will be the next research work.

## Figures and Tables

**Figure 1 sensors-23-09014-f001:**
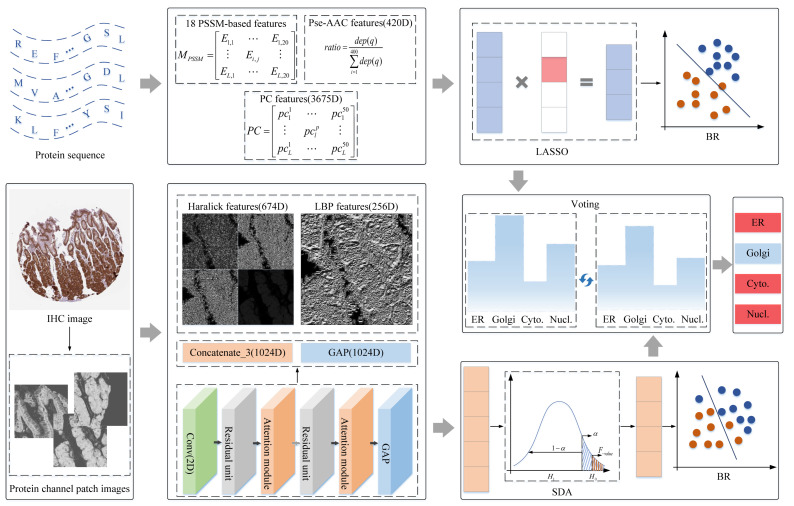
The flowchart of dual-signal model. Abbreviation definitions: ER: endoplasmic reticulum; Golgi: Golgi apparatus; Cyto.: cytosol; Nucl.: nucleoplasm.

**Figure 2 sensors-23-09014-f002:**
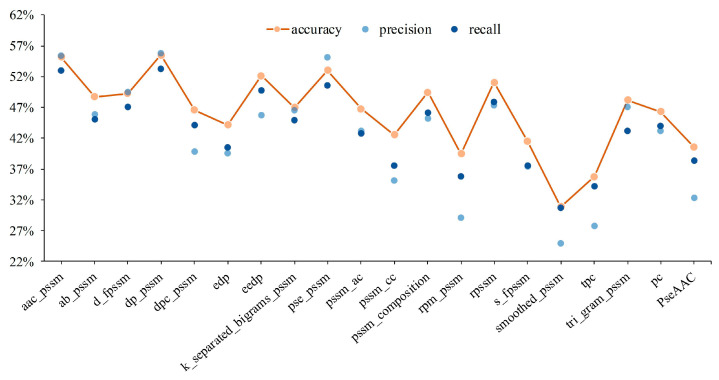
Experimental results for different shallow features of protein sequence.

**Figure 3 sensors-23-09014-f003:**
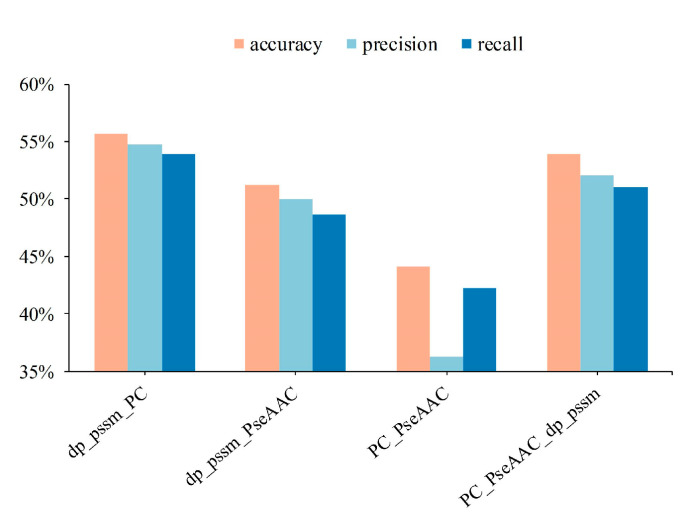
Performance of shallow features fusion of protein sequences.

**Figure 4 sensors-23-09014-f004:**
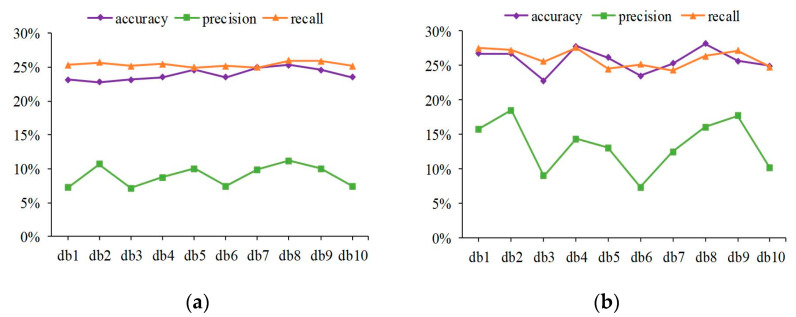
Experimental results of 10 dbs Daubechies filters in protein channel patch images. (**a**) Line plot of Haralick feature performance fluctuations in different Daubechies filters; (**b**) experimental results of Haralick and LBP concatenating.

**Figure 5 sensors-23-09014-f005:**
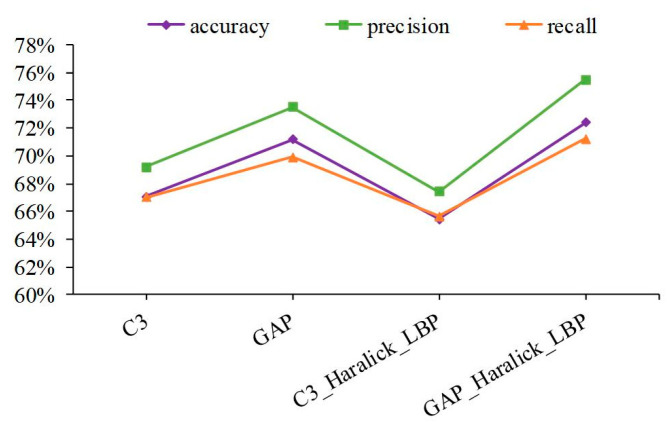
Performance comparison of IHC image in the abstract and concatenate features.

**Figure 6 sensors-23-09014-f006:**
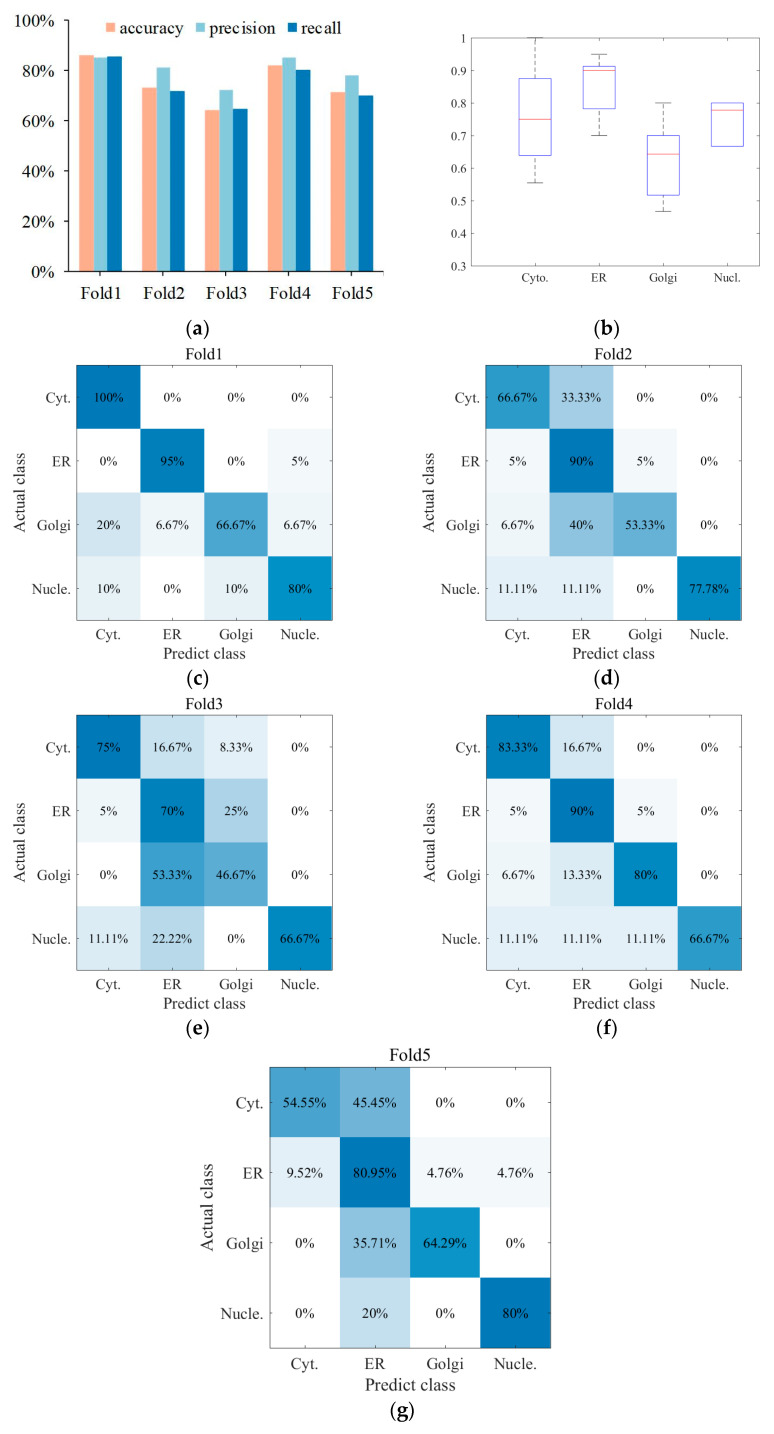
The multi-classifier outputs result and visualization by centralized voting mechanism. (**a**) Multi-classifier ensemble with five-fold cross-validation; (**b**) performance fluctuations of different protein subcellular locations in the five-fold cross-validation; (**c**–**g**) visualization of confusion matrix in four categories.

**Figure 7 sensors-23-09014-f007:**
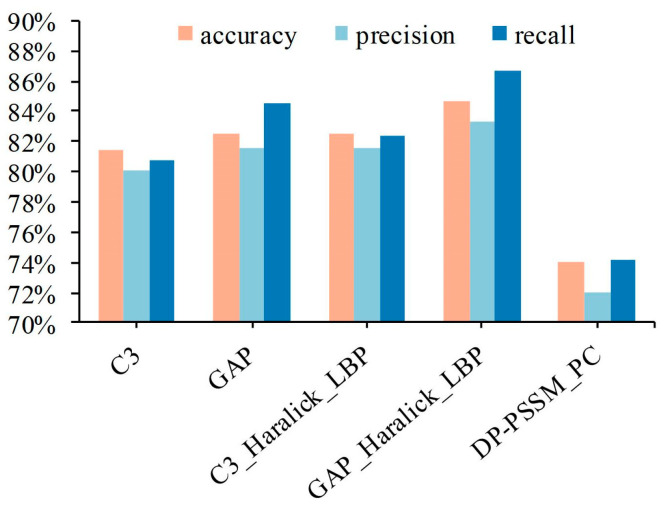
The experimental results of different feature space based on Leave-One-Out.

**Table 1 sensors-23-09014-t001:** The dataset distribution of protein sequences and IHC images.

Subcellular Location	Sequence	Image
ER	101	101
Golgi	74	74
Cytosol	59	59
Nucleoplasm	47	47
Total	281	281

## Data Availability

The data presented in this study are available on request from the corresponding author.

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
