# Peer review of "Dual-Signal Feature Spaces Map Protein Subcellular Locations Based on Immunohistochemistry Image and Protein Sequence"

_sensors, 2023, doi:10.3390/s23229014_

Round 1

Reviewer 1 Report

Comments and Suggestions for Authors

In this manuscript, Kai Zou et al. describe a dual-signal model by incorporating immunohistochemistry (IHC) images into protein sequences to predict protein subcellular locations. It's a good idea to concentrate on dual signals, which have two patterns instead of just one: 2-D picture patterns and 1-D sequence patterns. The authors construct a model to fit subcellular location distributions of proteins by adopting statistical operators, representation learning, and resemble learning. The benchmark dataset and experimental results have demonstrated the dual-signal model outperforms the single-signal model. However, I believe the authors could make a few improvements and provide further details. Below are some comments.

1. The benchmark dataset was collected based on the steps in section 2.1. The 287 samples account for a small fraction of the total 4772 samples involved. It is a significant quantity decrease.

2. Two dimensionality reduction methods were used for different protein signals as described in sections 2.2 and 2.3.

3. In section 3.3, some samples were more biased to other classifications. In addition to class-imbalance, does it reflect the unrobustness of the model? 

4. The language should be improved. Some details need polishing.

Reviewer 2 Report

Comments and Suggestions for Authors

The authors in this manuscript developed a dual-signal fusion system by incorporating IHC images into protein sequences to predict protein subcellular location, which can be accepted after addressing the following minor issue:

1.     In the abstract, the background is too long. Please give a brief introduction.

2.     The discussion part is short. Please add more related words.

3.     Please label the significant difference in Figure 2, 3 and 7.

4.     There are some errors in the text, please check carefully to revise them.

Comments on the Quality of English Language

Moderate editing of English language required.

Reviewer 3 Report

Comments and Suggestions for Authors

This paper titled “Dual-signal feature spaces map protein subcellular locations based on immunohistochemistry image and protein sequence” contributes to a dual-signal model of protein subcellular localization. From the experimental results, dual-signal can achieve better performance compared with image or sequence. It is a good case for multi-signal fusion in protein subcellular location. But, I think some clarification is necessary to ass to the experiment and conclusion. My detailed comments are as follows:

1.     As stated, the are some signals of protein, such as protein sequence, IHC images, and IF images, while protein sequence and IHC images are involved. The IF images show a clearer compartmental cellular structure than IHC images but received little attention.

2.     In section 3.3, the data bar, box plot, and confusion matrixes show good results, but rarely about the differences of every classification.

3.     In conclusion, it only expresses the model can reach 75.41%, 80.38%, and 74.38% in accuracy, precision, and recall respectively, but does not convey any signal advantage.

4.     The baseline experimental results were not presented.

5.     The English writing needs revise.

Comments on the Quality of English Language

Minor Editing needed.
